# Ecdysteroid responses to urban heat island conditions during development of the western black widow spider (*Latrodectus hesperus*)

**Claire Moen, J. Chadwick Johnson, Jennifer Hackney Price***

School of Math & Natural Sciences, Arizona State University—West Campus, Glendale, AZ, United States of America

* Jennifer.Hackney.1@asu.edu

## Abstract

The steroid hormone 20-hydroxyecdysone (20E) controls molting in arthropods. The timing of 20E production, and subsequent developmental transitions, is influenced by a variety of environmental factors including nutrition, photoperiod, and temperature, which is particularly relevant in the face of climate change. Environmental changes, combined with rapid urbanization, and the increasing prevalence of urban heat islands (UHI) have contributed to an overall decrease in biodiversity making it critical to understand how organisms respond to elevating global temperatures. Some arthropods, such as the Western black widow spider, *Latrodectus hesperus*, appear to thrive under UHI conditions, but the physiological mechanism underlying their success has not been explored. Here we examine the relationship between hemolymph 20E titers and spiderling development under non-urban desert (27˚C), intermediate (30˚C), and urban (33˚C) temperatures. We found that a presumptive molt-inducing 20E peak observed in spiders at non-urban desert temperatures was reduced and delayed at higher temperatures. Intermolt 20E titers were also significantly altered in spiders reared under UHI temperatures. Despite the apparent success of black widows in urban environments, we noted that, coincident with the effects on 20E, there were numerous negative effects of elevated temperatures on spiderling development. The differential effects of temperature on pre-molt and intermolt 20E titers suggest distinct hormonal mechanisms underlying the physiological, developmental, and behavioral response to heat, allowing spiders to better cope with urban environments.

## Introduction

Extreme temperature fluctuations associated with global climate change are exposing organisms to an unprecedented level of temperature stress [1–3]. Temperatures exceeding optimum can have multiple negative consequences including, but not limited to, immune system dysfunction [4], increased mortality rates [5–8], decreased fertility [9, 10], developmental delays

**Data Availability Statement:** All relevant data are within the paper and its Supporting Information files.

**Funding:** This work was supported by an ASU New College Undergraduate Interdisciplinary Research Experience (NCUIRE; https://newcollege.asu.edu/ncuire) award to CM, a Salt River Project Life Sciences Scholarship (https://scholarships.asu.edu/scholarship/94) to CM, a Central Arizona-Phoenix Long Term Ecological Research (CAP-LTER; https://sustainability-innovation.asu.edu/caplter/) award to CJ, and an ASU New College Scholarship, Research, and Creative Activities award to JHP and CJ. The funders had no role in study design, data collection and analysis, decision to publish, or preparation of the manuscript.

**Competing interests:** The authors have declared that no competing interests exist.

[7, 8], and slower growth rates [10–12]. Compounding the effects of climate change is the phenomenon of urbanization [10]. Approximately 55% of humans live in cities and that number is expected to increase to 68% by 2050 [13]. Urban development can affect organisms in several different ways. Urbanization causes habitat fragmentation or loss and the introduction of non-native species [14]. In addition, the Urban Heat Island (UHI) effect, which is caused by built structures (e.g. concrete and asphalt) within urban cities capturing heat during the day and retaining it throughout the night, results in warmer temperatures in developed areas in comparison to the surrounding rural environment [15, 16]. While the intensity of the thermal environment varies from city to city and between seasons and can be influenced by numerous factors such as 3D architecture and urban ventilation, elevated UHI temperatures and other factors associated with urbanization negatively affect many animals, contributing to an overall decrease in biodiversity in urban areas when compared to surrounding non-urban areas [17–26].

Some organisms, termed urban exploiters, thrive in urban environments, where they are often found at high densities in developed areas compared to surrounding nonurban regions [27]. Herring gulls (*Larus argentatus*), blackbirds (*Turdus merula*), house mice (*Mus musculus*), brown rats (*Rattus norvegicus*), and feral pigeons (*Columba livia*) are just some examples of vertebrate urban exploiters [28–34]. The effects of urbanization observed in vertebrates is mirrored in invertebrates, with urban populations being less diverse than those in rural environments while urban exploiters thrive in the face of urban change [35, 36]. For example, under urban conditions (increased temperature and decreased humidity), the wall brown butterfly (*Lasiommata megera*) demonstrates increased larval survival and the production of larger adults [37]. Similarly, urban humped golden orb-weaving spiders (*Nephila plumpipes*) are larger and produce more offspring than their rural counterparts [38].

One urban exploiter, the western black widow spider (*Latrodectus hesperus*), is flourishing in the face of urbanization [39, 40]. Black widows are found throughout the desert southwest including the Sonoran Desert and thrive in urban areas even though urban centers are significantly warmer than the surrounding non-urban desert [40]. Previous research has shown in the summer, during which females can produce several egg sacs, the average temperatures for arthropod microclimates in urban Phoenix, Arizona to be elevated by 6°C compared to average desert temperatures (urban = 33°C vs desert = 27°C; [7]). Even with these elevated urban temperatures, *L. hesperus* is found at densities that are 30 times more concentrated than their rural counterparts [40]. Despite this apparent success, we have recently shown that urban temperatures are detrimental to spider development [7]. Elevated temperatures increase spider mortality, increase the time between molts, and decrease growth rates leading to the production of smaller adults [7]. Spiders alter certain behaviors (e.g. decreased web building and increased aggression) under urban conditions, which, along with increased food availability, are thought to partially offset the negative effects of UHI temperatures [7, 40, 41]. However, understanding the physiological responses to urbanization may help us to better understand the success of these arachnids in urban environments in the face of the negative consequences of urban temperatures. Understanding the impact of rising temperatures on animal physiology will be of extreme importance as global temperatures continue to rise, with urban heat models being an early predictor of viability of organisms in the rapidly changing environment.

Molting in arthropods is regulated primarily by the steroid hormone 20-hydroxyecdysone (20E), which initiates gene expression cascades leading to the physiological, morphological, and behavioral changes associated with major developmental transitions such as molting, during which the larval cuticle is shed and replaced with a new, larger cuticle allowing for organism growth [42–49]. Ecdysteroid regulation of molting and metamorphosis has been extensively studied in a wide variety of insects including *Drosophila melanogaster, Bombyx*

*mori*, and *Manduca sexta* where a steep rise in the hemolymph ecdysteroid titer precedes each major developmental transition [49–56]. In contrast, while ecdysteroids have been identified in arachnids, relatively little research has been done to document ecdysteroid titers during spider development [57–61].

Modulation of hormone titers and signaling cascades are thought to modify the timing of development (e.g. time between transitions such as molts and metamorphosis) in response to environmental changes including fluctuations in temperature, photoperiod, population density, and nutrition [62–73]. For example, in the desert toad (*Scaphiopus couchii*), habitat desiccation is associated accelerated metamorphosis due to an increase in cortisol and thyroid hormone titers [74]. In the fruit fly (*Drosophila melanogaster*), nutritional shortages and elevated temperatures each lead to developmental arrests in egg development associated with increased circulating ecdysteroid titers [75–79]. Ectopic ecdysteroid administration in *Drosophila* suppresses egg development and egg laying [80, 81]. Together, these studies suggest that in adult flies, stress-induced ecdysteroid production may lead to physiological changes that shifts energy allocation from reproduction to survival [75–80]. Elevated ecdysteroid titers induced by thermal stress also regulates a subset of small heat shock proteins such as Hsp23 which is thought to play a neuroprotective role in response to stress [82].

Here we examine how temperatures associated with Urban Heat Islands influence ecdysteroid titers during early spider development. We reared spiders under three different temperatures that mimic non-urban desert, urban, and intermediate environments. Because UHI temperatures are associated with developmental delays, we hypothesized that heat would likely delay the surge of ecdysteroids that precedes and stimulates molting, hereafter referred to as the 'molting peak'. Because thermal stress is also associated with an overall increase in ecdysteroids in arthropods, we also hypothesized that high temperatures would result in basal ecdysteroid titers, which we refer to as the 'intermolt ecdysteroid titers', being increased throughout spider development. Our results confirm that UHI temperatures have negative effects on development (e.g. size, growth rate, mortality, time between molts) and are associated with changes in the ecdysteroid titers throughout spiderling development.

## Materials and methods

### Ethics statement

This study did not utilize animals requiring approval by an institutional review board or ethics committee. No field permits were involved. No other permissions were necessary for the research reported here.

### Sample collection

Adult female *L. hesperus* were collected from six collection sites across the urban Phoenix area (S1 Fig). Spiders were housed individually at 24˚C and fed one cricket on a weekly basis. For ten females, 100–250 eggs from their first egg sac were individually reared in 4.13 x 4.13 x 5.56 cm enclosures containing two toothpicks measuring 6.3 cm crossing diagonally to provide a structure for web building. For the first 30 days of development, eggs were reared at room temperature (24˚C) at a 12:12 photoperiod.

### Spider rearing

Starting 30 days after egg sac production each spider was fed two *Drosophila melanogaster* twice a week. Each spider was checked daily for molting and survival, which were recorded to track developmental progression, with shed cuticles being removed from enclosures when

found. On day 44 of development, which includes 30 days of incubation in the egg sac and 14 days post-hatch, families were divided into environmental chambers (Percival Scientific), simulating temperature conditions of interest, which included 27, 30 and 33˚C. Temperatures were chosen based on previous studies that demonstrated that in the summer, the average temperatures for arthropod microclimates in urban Phoenix, Arizona is elevated by 6˚C compared to average desert temperatures (urban = 33˚C vs desert = 27˚C; [7]). Before day 44 of development, spiderlings transferred to higher temperatures after being removed from the egg sac do not typically survive (C. Moen, unpublished observation). In addition, inconsistencies were observed in the timing of molts prior to day 44 of development, when competition is highest between peers. Developmental data [molts, deaths, and mass] were recorded daily until day 75 of development. Data was not collected after day 75 of development as differences between males and females become noticeably different after day 75 (C. Moen, unpublished observation). Due to the addition of a $3^{rd}$, intermediate temperature (30˚C) during the second year of this study, only 6 of the 10 spider families were subjected to this temperature.

## Hormone extraction

Spiders were individually weighed and preserved in 200 μL of methanol. Samples were homogenized using plastic pestles and centrifuged for 20 minutes at 18,000 x g. Supernatants were collected while remaining insoluble material was again homogenized and centrifuged. The resulting supernatants were combined and dried using an Eppendorf Vacufuge. Dried samples (hormone extracts) were resuspended in 200 μL of EIA Buffer (0.4M NaCl, 1 mM EDTA, 0.1% BSA in 0.1M phosphate buffer), enough for two replicates to be carried out per spider.

## Measurement of hormone concentrations

Ecdysteroid concentrations were quantified using a competitive EIA (enzyme immunoassay) kit (Cayman Chemicals, Inc., USA) according to manufacturer's instructions. In this assay, 20-hydroxyecdysone (20E) and 20E acetylcholinesterase (AChe), which were used as the standard and tracer respectively, compete for a limited number of binding sites on a rabbit anti-20E antibody. Ellman's Reagent was used as the substrate, which was converted to a yellow product by AChe. Therefore, the production of product is inversely proportional to the amount of 20E in the sample. Standard curves were prepared with the 20E EIA Ecdysteroid Standard (Cayman Chemicals, Inc. USA) using concentrations ranging from 10 to 0.020 ng/μl. Because the antibody detects 20-hydroxyecdysone and its' metabolites, we report 20E concentrations as 20E equivalents per mg tissue. The absorbance of samples and standards was measured at 415 nm using an ELX801U Ultra Microplate Reader (Bio-Tek Instruments). Microsoft Excel was used to analyze all resulting data. Absorbance data were linearized using a Logit transformation according to the following formula: $(B/B_0) = ln[B/B_0/(1-B/B_0)]$, where B represents the absorbance of samples or standards and $B_0$ represents maximum binding, the absorbance obtained in samples containing all assay components except 20E. The standard curve was generated by plotting the logit-transformed data vs. log concentrations and linear regression analysis was used to determine the concentration of each spider sample. Resulting ecdysteroid concentrations were converted to pg 20E Equivalents/mg tissue to account for the mass of each spider. Ecdysteroid concentrations from 9 spiderlings did not fall within the limits of the 20E standard curve were not included in any subsequent analyses.

## Statistics

Data were analyzed via a One-Way ANOVA performed using the ANOVA Shiny web application available at istats.shinyapps.io/ANOVA. Shiny apps are interactive online data analysis

and visualization tools that are created using Rstudio's Shiny framework [83]. Pairwise comparisons were calculated using the Tukey multiple comparison test. Results were considered statistically significant if $p \leq 0.05$ and marginally significant if $p < 0.1$. Figures were assembled in PowerPoint (Microsoft) and the GNU Image Manipulation Program (GIMP v2.10.10). Unless otherwise indicated, statistical data are presented as mean ± SEM.

## Results

### Assessment of ecdysteroids during development at different temperatures

There was a significant effect of temperature on average ecdysteroid titers during development of Western black widow (*L. hesperus*) spiderlings (27˚C: desert = 302 ± 24 vs 30˚C: intermediate = 259 ± 17 vs 33˚C: urban = 355 ± 29 pg 20E/mg; $F_{2,211} = 3.963$, $p = 0.0204$; See Fig 1). Ecdysteroid titers at intermediate temperatures were similar to those measured at desert temperatures. However, urban temperatures were associated with significantly higher 20E titers than intermediate temperatures ($T_{140} = -2.82$, $p = 0.01$). While we detected a significant effect of spider family on average 20E titers at desert temperatures ($F_{9,103} = 4.13$, $p = 0.0001$; S2A Fig), ecdysteroid titers in all families appear to respond in a similar fashion to elevated temperatures. With few exceptions, relative to desert temperatures (27˚C), average ecdysteroid titers in each family were decreased at intermediate temperatures (30˚C) and increased at urban temperatures (33˚C) (S2B Fig).

Analysis of daily changes in ecdysteroid titers revealed that at 27˚C, a sharp increase in ecdysteroid titers occurred two days prior to the second molt (Fig 2A). This large ecdysteroid peak was absent in spiderlings reared at 30˚C (Fig 2B) and 33˚C (Fig 2C). One day after the second molt, spiderlings from 27˚C and 30˚C treatments both exhibited a small pulse of ecdysteroids (Fig 2A and 2B). Additional small ecdysteroid peaks were visible in spiderlings at 30˚C following the second molt, but these did not occur at the same time as those observed in spiderlings from the 27˚C desert temperatures (Fig 2A).

Because increased temperatures are known to alter the timing of the second molt [7], it is possible that pre-molt ecdysteroid peaks were obscured when spiderlings reared at 30˚C and 33˚C were normalized to molts occurring at 27˚C. We therefore examined ecdysteroid developmental profiles in spiderlings staged based on the time at which the second molt occurred for each family at each temperature (Fig 3). Like what was observed at 27˚C, this method revealed a small ecdysteroid peak occurring two days prior to the second molt in spiderlings reared at 30˚C (Arrowhead; Fig 3A and 3B). In contrast, the pre-molt peak was not detected in spiderlings reared at 33˚C (Arrowhead; Fig 3C).

Analysis of ecdysteroid titers measured during two time intervals, one during the peak observed two days before the second molt (Fig 4, Molt) and the second spanning 2–5 days following the second mole (Fig 4, Intermolt), revealed a significant interaction between temperature and developmental phase ($F_{2,1} = 10.723$, $p = 0.025$). The molting peak at 27˚C was significantly higher than that observed at both 30˚C ($T_4 = 5.52$, $p = 0.000$). and 33˚C ($T_5 = 6.85$, $p = 0.000$). No significant difference was observed between the molting ecdysteroid peak at 30˚C and 33˚C. In contrast, basal titers were significantly higher at 33˚C when compared to 27˚C ($T_{53} = -2.68$, $p = 0.02$) and 30˚C ($T_{53} = -3.18$, $p = 0.01$).

### Assessment of developmental changes in response to various temperatures

To better understand how different temperatures influence development, which is regulated by ecdysteroids in arthropods and arachnids, we analyzed several developmental metrics: timing of the second molt, growth rate, predicted size at the second molt, and mortality (Fig 5). Consistent with previous reports [7], we determined that spiderlings reared at 33˚C underwent

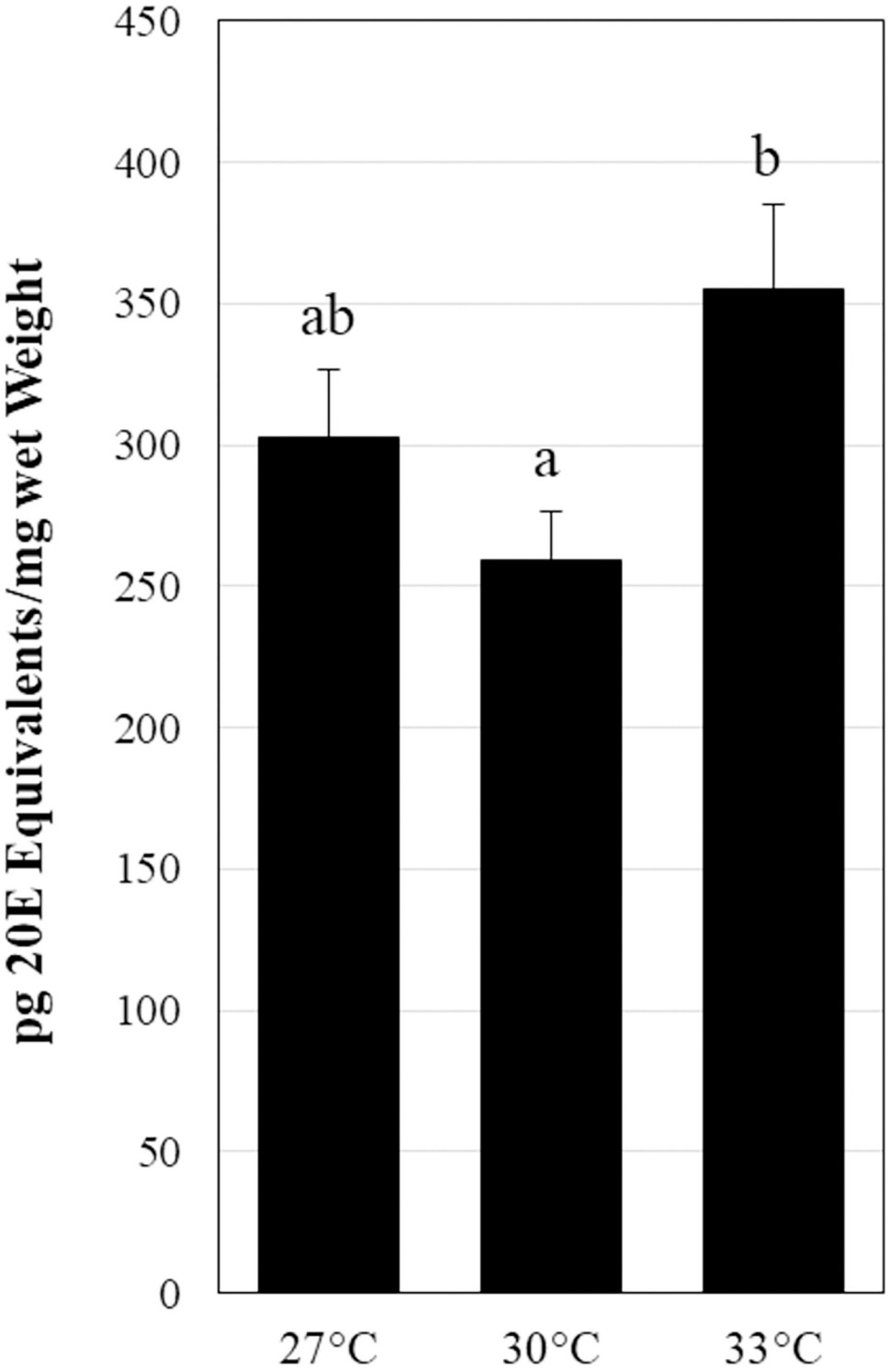

**Fig 1. Average ecdysteroid titers during spiderling development.** Ecdysteroid titers were determined for spiderlings reared at 27, 30, or 33˚C from 4 days before to 10 days after the second molt that occurred per family at 27˚C. Error bars represent standard error.

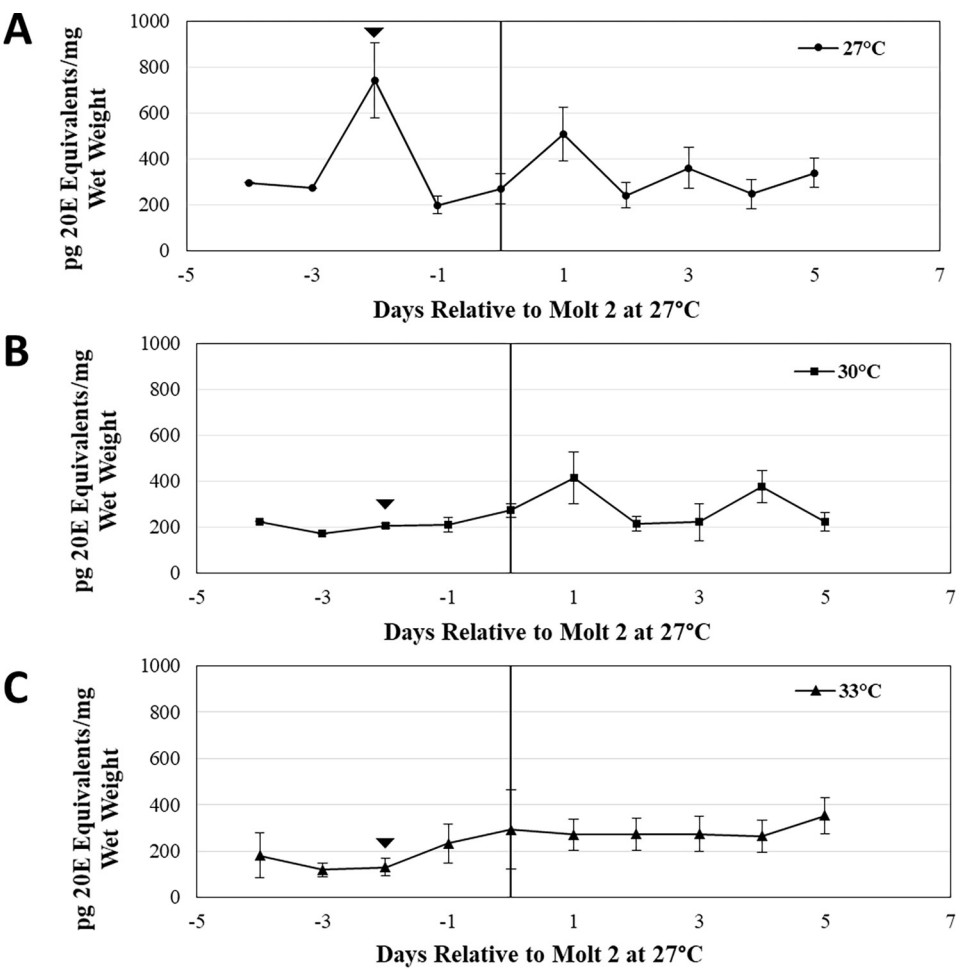

**Fig 2. Developmental ecdysteroid profiles at different temperatures.** (A-C) Average daily ecdysteroid titers from spiderlings reared at 27˚C (A), 30˚C (B), and 33˚C (C). Only families that had spiderlings reared at both temperatures were used for analysis. Spiderling ages are relative to the timing of the second molt for each family, at 27˚C. Error bars represent standard error.

the second molt approximately three days later than siblings reared at 27˚C and at 30˚C (Fig 5A). However, the effects of temperature on the timing of the second molt were not statistically significant.

There was a significant effect of temperature treatments on spiderling growth rate ($F_{2,15}$ = 5.191, p = 0.0194; Fig 5B). Spiders at 33˚C grew significantly slower than siblings reared at 27˚C ($T_{10}$ = 3.02, p = 0.02) and marginally slower than 30˚C ($T_{10}$ = 2.49, p = 0.06).

In some arthropods, organisms much reach a minimum size before molting will begin [70]. We therefore examined the effects of temperature on spider mass and found that temperature had a significant effect on the size of spiders at the time of the second molt (Fig 5C; $F_{2,15}$ = 4.284; $p$ = 0.0337). While there was no significant difference between the predicted size for spiders housed at 27˚C (4.37 ± 0.68 mg) and 30˚C (3.99 ± 0.53 mg) at the time of the second molt, spiders reared at 33˚C were significantly smaller (1.91 ± 0.67 mg) at the time of molt two than those reared at 27˚C ($T_{10}$ = 2.72, p = 0.04) and marginally smaller than those at 30˚C ($T_{10}$ = 2.30, p = 0.09; Fig 5C).

There was a significant effect of temperature on mortality (Fig 5D; $F_{2,15}$ = 4.103; $p$ = 0.0379). At 33˚C, 11.0% ± 4.8% of spiders died during the course of the study, which was a

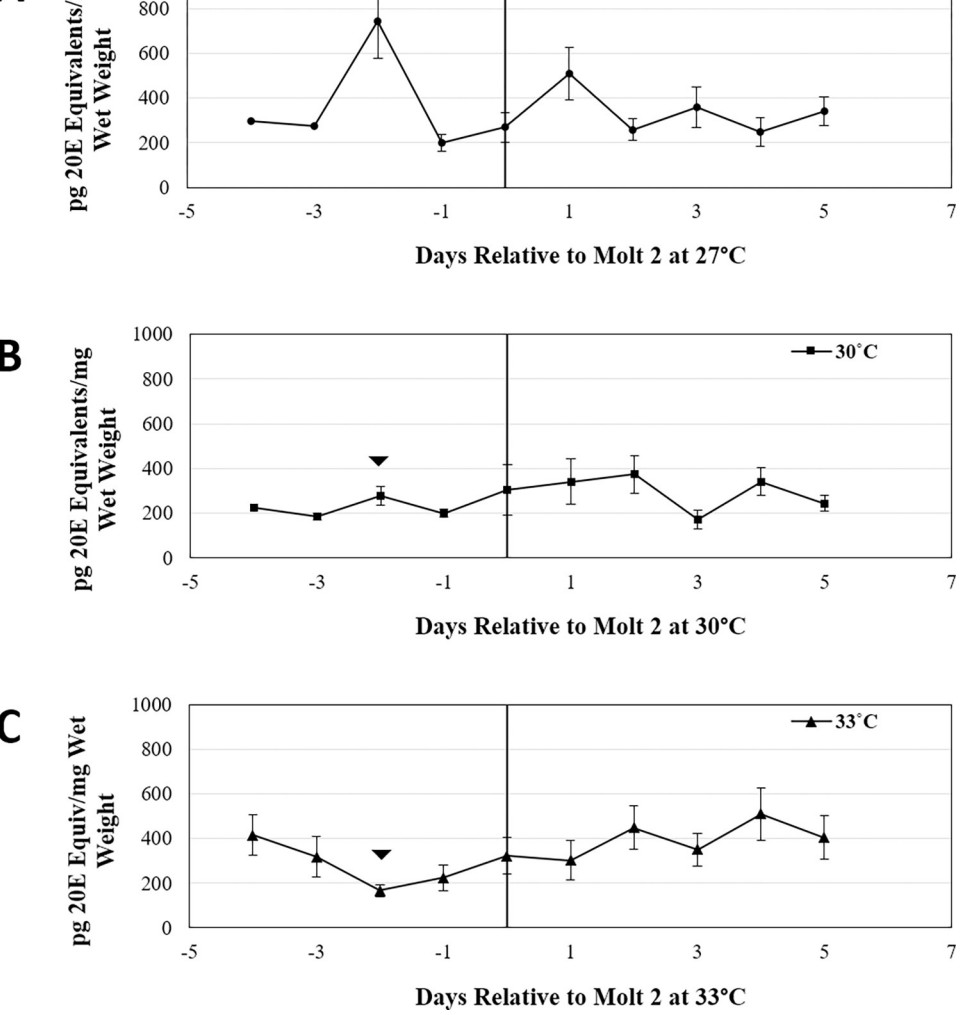

**Fig 3. Normalized developmental ecdysteroid profiles.** Average daily ecdysteroid titers from spiderlings reared at (A) 27°C, (B) 30°C and (C) 33°C. Only families that had spiderlings reared at all three temperatures were used (N = 6 families). Spiderling ages are relative to the timing of the second molt for each family, at each temperature. The presumptive molt-inducing peak of ecdysone is indicated (arrowhead). Error bars represent standard error.

significantly higher mortality rate than siblings reared at 27°C ($T_{10}$ = -2.71, p = 0.04), of which only 4.3% ± 3.2% spiders did not survive the study (Fig 5D). No statistically significant differences in mortality rates were observed between spiders reared 30°C (5.6% ± 4.6%) when compared to those at 27°C, or 33°C.

## Discussion

As reported here and in Johnson et al. [7], despite the apparent success of *L. hesperus* in urban Phoenix, UHI conditions are correlated with decreased growth rates, delayed development, and increased mortality. We have found that these heat-induced developmental changes are associated with underlying changes in ecdysteroid titers, which respond to increased temperatures in two ways. First, slightly elevated temperatures, from 27 to 30°C, result in a molting peak that is delayed and reduced in size, while increasing the temperature to UHI conditions

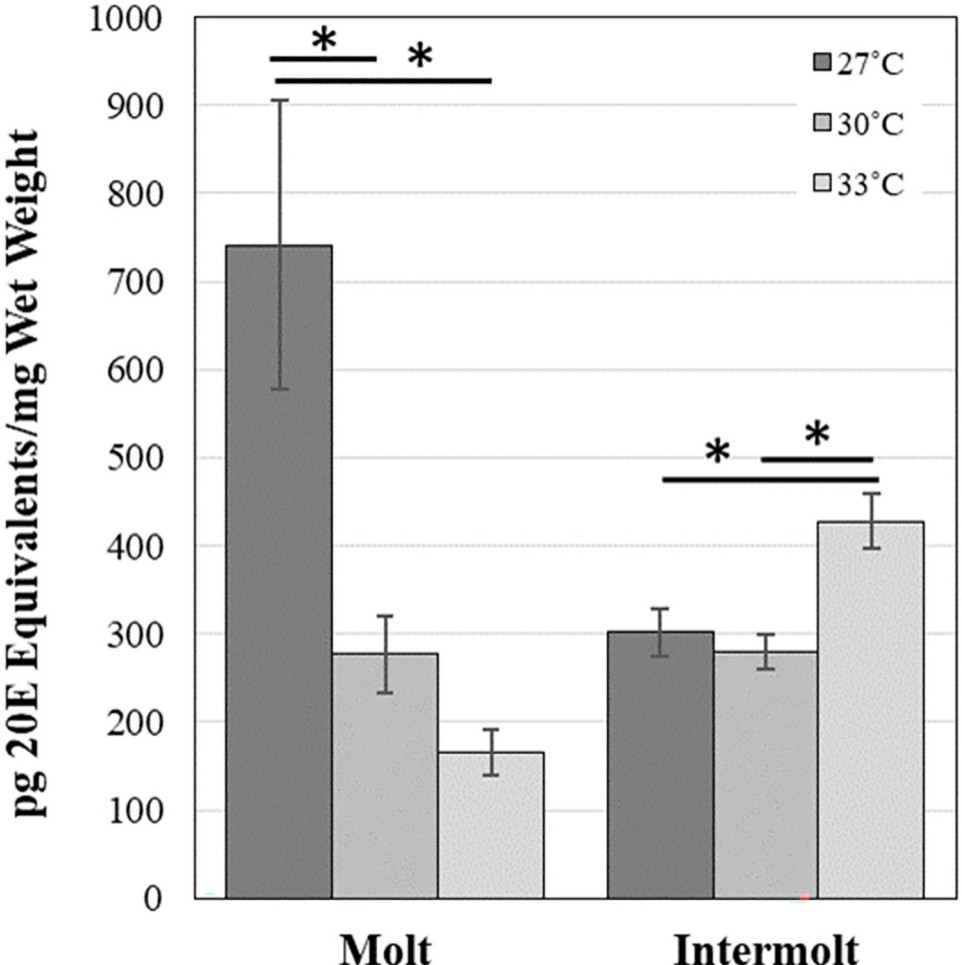

**Fig 4. Phase-specific ecdysteroid titers.** Average daily ecdysteroid titers from spiderlings reared at 27˚C, 30˚C and 33˚C were determined at two time points: two days prior to the second molt (Molt) and days 2–5 after the second molt (Intermolt). Only families that had spiderlings reared at all three temperatures were used (N = 6 families). Spiderlings were aged relative to the timing of the second molt for each family, at each temperature. Error bars represent standard error. (*) indicates a significant difference (p<0.05).

(33˚C) appears to completely abolish the molting peak. The second ecdysteroid response to elevated temperatures occurs during the intermolt period between the second and third molts. During this time, raising the temperature to intermediate temperatures (30˚C) does not alter the basal ecdysteroid titers. However, the intermolt ecdysteroid titers are significantly elevated in response to UHI temperatures (33˚C).

Developmental timing (e.g. time between molts), which is regulated by ecdysteroids, is highly dependent upon temperature in arthropods [84]. Previous reports, as well as the present study, have indicated that there is an optimum temperature for development and temperatures that stray too far outside of this range have negative effects on arthropod growth, developmental times, and mortality [7, 85]. We have observed this response in *L. hesperus* and, although the spiders utilized in this study originated from urban populations, their optimum temperature for development appears to be 27˚C, the average temperature of spider microclimates in the non-urban Sonoran Desert [7]. At this temperature, a surge of ecdysteroids initiates molting, and only low levels of spider mortality is observed. As temperatures move away from this

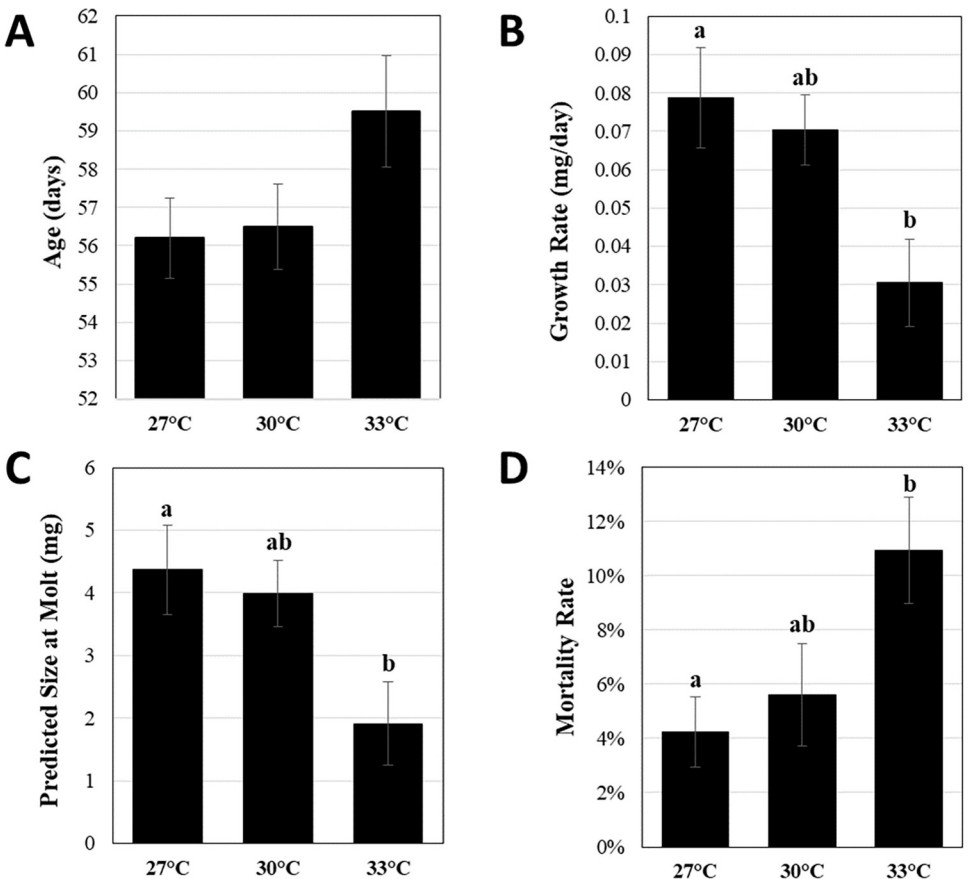

**Fig 5. Effects of elevated temperatures on development.** (A) Average molt times for spiderlings reared at different temperatures. (B) Average growth rates were measured from 55–75 days of development for spiderlings reared at different temperatures. (N = 6 families per treatment) (C) Average spiderling size at time of second molt was predicted from molt times and growth rates for each family (N = 6 families per treatment). (D) Spiderling mortality at each temperature was determined from 55–75 days of development for each family. (A-D) Different lowercase letters indicate significant differences (p<0.05). N = 6 families per treatment.

optimum, from 27 to 30˚C, the 20E surge is reduced and delayed, although these changes do not appear to have a significant effect on molting times, growth rates, or mortality. In contrast, raising temperatures to 33˚C, the average summer microclimate temperatures experienced by urban spiders, the ecdysteroid surge is reduced or missing and spiders exhibit significant molting delays, slower growth rates, and increased spider mortality. Although our results do not take into consideration daily fluctuations in temperature, a variable that must be addressed in future studies, our data suggest that 33˚C exceeds the temperature threshold for development and any spiders that are able survive to adulthood will be smaller than their counterparts in the non-urban desert.

In various arthropods, short-term exposure to temperatures above the optimum developmental threshold can lead to a delay in development, with recovery possible after returning to normal conditions [86–88]. However, UHI conditions are not temporary and studies with Sonoran *Drosophila* species suggest that, at least in this case, microhabitats provide little-to-no respite from extreme temperatures [89]. Results presented here suggest that temperatures associated with urban heat islands represent chronic thermal stress that disrupts the ecdysteroid surge that is essential for molting, resulting in overall negative effects on spider development. The possibility that a return to lower temperatures could restore the normal developmental

profile, both in terms of ecdysteroid titers and developmental timing, is intriguing, but is outside the scope of the present study.

In many arthropods, formation of the molt-inducing ecdysteroid peak occurs in response to the attainment of a critical weight, a minimal mass which indicates that sufficient nutritional stores have been attained to support the upcoming molt [47, 70, 90]. While critical weight has not been reported in spiders, the presence of a size-sensing mechanism that triggers ecdysteroid production would further explain why the pre-molt ecdysteroid peak is delayed in spiders reared at high temperatures, which have significantly slower growth rates than their cooler counterparts.

A small increase in basal ecdysteroid levels has been reported in arthropods reared under stressful conditions including chronic intoxication, nutrient shortage, oxidative stress, sleep deprivation, high population density, injury, and thermal stress [77, 91–94]. A similar endocrine response to elevated temperatures has been studied extensively in *Drosophila melanogaster*, with an increase in ecdysteroid levels being seen within 60 minutes of exposure to heat stress and, while not attaining the same intensity of the morphogenic peak, intermolt hemolymph ecdysteroid titers are significantly elevated compared to non-stressed controls [77, 94].

Published reports suggest that elevated basal 20E titers may serve a protective role during stressful conditions, but only if the concentrations are below those that induce molting and metamorphosis [95–97]. Moderately elevated ecdysteroid titers are associated with increased resistance to formaldehyde in the blue bottle fly *Calliphora vicina* and in the silkworm *Bombyx mori* [95]. In fruit flies (*Drosophila melanogaster)* and the greater wax moth (*Galleria mellonella*), moderately elevated basal 20E titers are required for the regeneration of damaged tissues while high 20E concentrations induce developmental transitions (e.g. pupation) but suppress tissue regeneration [96, 97]. Protective effects of increased basal ecdysteroid titers have also been reported following bacterial infection, where moderate 20E titers reduce mortality associated with paralyzing toxins and septicemia induced by spore infection [91].

Elevated intermolt ecdysteroid concentrations induced in spiders exposed to extreme heat might serve a thermoprotective role by altering physiological and behavioral responses so that any animals that are able to survive to adulthood can better cope with the unfavorable urban environment. Increased tolerance to high temperatures in city-dwelling arthropods has been reported for urban leaf-cutter ants, which are more thermotolerant than their rural counterparts [98]. While ecdysteroid levels in heat tolerant ants has not been examined, ecdysteroid have been linked to thermotolerance in other systems [99, 100]. Ecdysteroids can induce expression of various genes encoding heat shock proteins (Hsps), even in the absence of heat [99, 100]. Heat shock proteins are chaperones that prevent the aggregation of denatured proteins and are associated with resistance to heat shock and other forms of stress [101, 102]. Enhanced expression of Hsps, specifically *Hsp30* and *Hsp70*, in desert species of the small freshwater fish *Poeciliopsis* is thought to confer thermal resistance, allowing the fish to survive elevated temperatures found in desert regions of northwest Mexico [103, 104]. Similarly, increased expression of Hsp genes including *Hsp90* and *Hsp47* confers thermal resistance to a population of minnows (*Puntius sopher*) found in hot spring run-offs in Odisha, India [105]. Increased thermotolerance has also been reported in cultured cells that overexpress *Hsp70* or *Hsp27* [106–109]. Although Hsps that are induced in response to thermal stress have been identified in the wolf spider, *Pardosa pseudoannulata*, ecdysteroid regulation of Hsps in spiders has not been established [110].

Elevated basal ecdysteroid titers might also mediate behavioral changes in web-building and aggression that have been reported in urban spiders [7]. Urban males show increased aggression towards prey when compared to males at lower non-urban desert temperatures while juvenile urban females have reduced web building behavior and produce smaller webs

[7]. A link has been shown between ecydsteroid concentrations and web-building behavior in the orb-weaver spiders, *Cyclosa morretes* and *Cyclosa fililineata* [111]. These spiders serve as hosts for parasitoid wasps, which manipulate web-building behavior of their spider hosts by injecting ecdysteroids [111]. Parasitized orb-weaver spiders display elevated ecdysteroid titers and build 'cocoon webs' which are thought to protect the wasp's cocoon as it develops [111]. A link between ecdysteroids and aggressive behavior has not been examined in spiders but has been demonstrated in other organisms. In honeybees, dominant workers had significant higher ecdysteroid titers than lower-ranked workers and presented with higher levels of aggression [112]. In American lobsters, *Homarus americanus*, elevated ecdysteroid titers have also been linked to increased aggression [113–115]. Pre-molt lobsters have high levels of 20E and are more aggressive than lobsters at different developmental stages [114]. In addition, lobsters injected with 20E also display an increase in aggressive behavior [115].

It should be noted that genomic work suggests large genetic variation between urban Phoenix and nearby Sonoran Desert black widow populations [116]. The present study has focused solely on responses in urban spiders; however, the possibility exists that spiders from desert lineages would respond differently to UHI temperatures. A natural extension of our work would be to compare ecdysteroid titers, development, and behavior in urban and desert spider lineages reared at both desert (27˚C) and urban (33˚C) temperatures. Indeed, we have recently shown an intriguing interaction in which desert lineages are more cannibalistic than urban ones, but the UHI temperature of 33˚C makes all spiderlings heighten cannibalism [117].

## Conclusion

Despite the apparent success of *L. hesperus* as an urban exploiter, we find that elevated urban temperatures significantly delay and reduce the ecdysteroid peak that precedes the second molt, a finding that is consistent with the developmental retardation observed in urban spiders [7]. Our studies suggest that current urban temperatures permit limited survival and moving forward, *L. hesperus* with less susceptibility to environmental extremes will likely be selected for as temperatures continue to rise. We also find that urban temperatures are associated with significantly elevated intermolt ecdysteroid titers which may, at least in part, counteract the negative effects on developmental progression. The potential thermoprotective effects of elevated basal ecdysteroids, ecdysteroid mediated changes in spider behavior, and other factors such as increased prey abundance [118] and temperature variation within a 24-hour period, may serve to counteract some of the negative consequences of increased temperatures, allowing black widows to continue exploiting urban settings.

## Supporting information

**S1 Fig. *L. hesperus* collection sites in the Phoenix metropolitan area.** Samples were collected from the following sites FLW (Frank Lloyd Wright), MR (Marshall Ranch), SBF (Sun Burst Farms), OLI (Olive), AVOW (Avondale West—Wigwam Creek Middle School), and AVOC (Corte Sierra Middle School). Spider families were named based on collection site.
(TIF)

**S2 Fig. Variation in family ecdysteroid titers during spiderling development.** (A) A significant effect of family on average ecdysteroid titers was observed at 27˚C ($F_{9,103}$ = 4.13; p = 0.0001). (B) Most families had lower ecdysteroid titers at intermediate temperatures and higher ecdysteroid titers at urban temperatures when compared to titers observed at desert temperatures.
(TIF)

**S1 Table. Spider ecdysteroid and developmental measurements.**
(DOCX)

# Acknowledgments

We would like to thank the following lab members who contributed to this study: Abigail Murray, Edgar Aragon, Gabbie Minehan, Keaton Coker, Ryan Clark, Javi Urcuyo, Alicia Sandoval, Emily Garver, Taylor Martin, Sofia Vine, Giselle Gregorio, Ahmed Alizzy, and Dale Stevens.

# Author Contributions

**Conceptualization:** J. Chadwick Johnson, Jennifer Hackney Price.

**Formal analysis:** Jennifer Hackney Price.

**Funding acquisition:** Claire Moen, J. Chadwick Johnson, Jennifer Hackney Price.

**Investigation:** Claire Moen.

**Methodology:** Claire Moen, J. Chadwick Johnson, Jennifer Hackney Price.

**Project administration:** J. Chadwick Johnson, Jennifer Hackney Price.

**Resources:** J. Chadwick Johnson, Jennifer Hackney Price.

**Supervision:** J. Chadwick Johnson, Jennifer Hackney Price.

**Writing – original draft:** Jennifer Hackney Price.

**Writing – review & editing:** Claire Moen, J. Chadwick Johnson, Jennifer Hackney Price.

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
