## [Decision Letter · Decision Letter 0]

7 Feb 2022

PONE-D-21-35850Ecdysteroid Responses to Urban Heat Island Conditions during Development of the Western Black Widow Spider (Latrodectus hesperus)PLOS ONE

Dear Dr. Hackney Price,

Thank you for submitting your manuscript to PLOS ONE. After careful consideration, we feel that it has merit but does not fully meet PLOS ONE’s publication criteria as it currently stands. Therefore, we invite you to submit a revised version of the manuscript that addresses the points raised during the review process.

We look forward to receiving your revised manuscript.

Kind regards,

Jun Yang

Academic Editor

PLOS ONE

Journal Requirements:

“This work was supported by an ASU New College Undergraduate Interdisciplinary Research Experience (NCUIRE; https://newcollege.asu.edu/ncuire) award to CM, a Salt River Project Life Sciences Scholarship (https://scholarships.asu.edu/scholarship/94) to CM, a Central Arizona-Phoenix Long Term Ecological Research (CAP-LTER; https://sustainability-innovation.asu.edu/caplter/) award to CJ, and an ASU New College Scholarship, Research, and Creative Activities award to JHP and CJ.”

Additional Editor Comments:

Reviewer 1

The authors explored the responses to temperature during development of the Western Black Widow Spider. The research methodologies are reasonable, and the findings are interesting. However, there are still a few aspects that should be improved to make the paper publishable. The paper is not well-organized. I focus here only on some points, which are hopefully easy for the authors to take into account in the revision.

1. Introduction- The author introduced the negative impacts of UHI. However, relevant research on UHI is lacking. Some important relevant references should be cited as follows.

Spatial Variability and Temporal Heterogeneity of Surface Urban Heat Island Patterns and the Suitability of Local Climate Zones for Land Surface Temperature Characterization. DOI: 10.3390/rs13214338.

Influence of urban morphological characteristics on thermal environment, Sustainable Cities and Society (2021), https://doi.org/10.1016/j.scs.2021.103045.

Contribution of urban ventilation to the thermal environment and urban energy demand: Different climate background perspectives, Science of the Total Environment (2021), https://doi.org/10.1016/j.scitotenv.2021.148791.

Suitability of human settlements in mountainous areas from the perspective of ventilation: a case study of the main urban area of Chongqing, Journal of Cleaner Production (2021), https://doi.org/10.1016/j.jclepro.2021.127467.

The impact of urban renewal on land surface temperature changes: A case study in the main city of Guangzhou, China. Remote Sensing (2020), https://doi.org/10.3390/rs12050794.

Impacts of Neighboring Buildings on the Cold Island Effect of Central Parks: A Case Study of Beijing, China. Sustainability (2020), doi: 10.3390/su12229499.

2. The author emphasizes UHI, while temperature was chosen in this manuscript. 27 ˚C, 33˚C and 37 ˚C can represent UHI? It is more appropriate to explore the effects of temperature.

3. Figure 2 is not clear enough, please modify it.

4. There is no Conclusion in this manuscript, please confirm it.

Reviewer 2

After a very careful reading of the work entitled "Ecdysteroid Responses to Urban Heat Island Conditions during Development of the Western Black Widow Spider", I have found a very well-done work, well presented, and organized, clear in concepts and methodology. The topic and context attract attention for many readers from various disciplines. The study is worth to be published after conducting the revisions.

I just suggest the author(s) add uncertainty analysis to enhance the scientific rigor of the research.

Reviewers' comments:

Reviewer's Responses to Questions

**Comments to the Author**

1. Is the manuscript technically sound, and do the data support the conclusions?

Reviewer #1: Yes

Reviewer #2: Yes

2. Has the statistical analysis been performed appropriately and rigorously? 

Reviewer #1: Yes

Reviewer #2: Yes

3. Have the authors made all data underlying the findings in their manuscript fully available?

Reviewer #1: Yes

Reviewer #2: Yes

4. Is the manuscript presented in an intelligible fashion and written in standard English?

Reviewer #1: Yes

Reviewer #2: Yes

5. Review Comments to the Author

Reviewer #1: The authors explored the responses to temperature during development of the Western Black Widow Spider. The research methodologies are reasonable, and the findings are interesting. However, there are still a few aspects that should be improved to make the paper publishable. The paper is not well-organized. I focus here only on some points, which are hopefully easy for the authors to take into account in the revision.

1. Introduction- The author introduced the negative impacts of UHI. However, relevant research on UHI is lacking. Some important relevant references should be cited as follows.

1) Spatial Variability and Temporal Heterogeneity of Surface Urban Heat Island Patterns and the Suitability of Local Climate Zones for Land Surface Temperature Characterization. DOI: 10.3390/rs13214338.

2) Influence of urban morphological characteristics on thermal environment, Sustainable Cities and Society (2021), https://doi.org/10.1016/j.scs.2021.103045.

3) Spatial Evolution of Population Change in Northeast China During 1992–2018, Science of the Total Environment (2021), https://doi.org/10.1016/j.scitotenv.2021.146023.

4) Contribution of urban ventilation to the thermal environment and urban energy demand: Different climate background perspectives, Science of the Total Environment (2021), https://doi.org/10.1016/j.scitotenv.2021.148791.

5) Suitability of human settlements in mountainous areas from the perspective of ventilation: a case study of the main urban area of Chongqing, Journal of Cleaner Production (2021), https://doi.org/10.1016/j.jclepro.2021.127467.

6) Optimizing local climate zones to mitigate urban heat island effect in human settlements, Journal of Cleaner Production (2020), https://doi.org/10.1016/j.jclepro.2020.123767.

7) The impact of urban renewal on land surface temperature changes: A case study in the main city of Guangzhou, China. Remote Sensing (2020), https://doi.org/10.3390/rs12050794.

8) Impacts of Neighboring Buildings on the Cold Island Effect of Central Parks: A Case Study of Beijing, China. Sustainability (2020), doi: 10.3390/su12229499.

2. The author emphasizes UHI, while temperature was chosen in this manuscript. 27 ˚C, 33˚C and 37 ˚C can represent UHI? It is more appropriate to explore the effects of temperature.

3. Figure 2 is not clear enough, please modify it.

4. There is no Conclusion in this manuscript, please confirm it.

Reviewer #2: After a very careful reading of the work entitled "Ecdysteroid Responses to Urban Heat Island Conditions during Development of the Western Black Widow Spider", I have found a very well-done work, well presented, and organized, clear in concepts and methodology. The topic and context attract attention for many readers from various disciplines. The study is worth to be published after conducting the revisions.

I just suggest the author(s) add uncertainty analysis to enhance the scientific rigor of the research.

6. PLOS authors have the option to publish the peer review history of their article (what does this mean?). If published, this will include your full peer review and any attached files.

Reviewer #1: No

Reviewer #2: No

---

## [Author Response · Author response to Decision Letter 0]

25 Mar 2022

Response to Reviewers – List of Responses

Journal Requirements:

The manuscript has been reformatted to meet journal style requirements.

“This work was supported by an ASU New College Undergraduate Interdisciplinary Research Experience (NCUIRE; https://newcollege.asu.edu/ncuire) award to CM, a Salt River Project Life Sciences Scholarship (https://scholarships.asu.edu/scholarship/94) to CM, a Central Arizona-Phoenix Long Term Ecological Research (CAP-LTER; https://sustainability-innovation.asu.edu/caplter/) award to CJ, and an ASU New College Scholarship, Research, and Creative Activities award to JHP and CJ.”

 The minimal data set has been included as a Supporting Information File: 

 S1 Table. Spider ecdysteroid and developmental measurements

The reference list in our manuscript is complete and accurate and has been cross-checked against the Retraction Watch Database. The reference list does not include any retracted articles. 

Additional Editor Comments:

Reviewer 1

The authors explored the responses to temperature during development of the Western Black Widow Spider. The research methodologies are reasonable, and the findings are interesting. However, there are still a few aspects that should be improved to make the paper publishable. The paper is not well-organized. I focus here only on some points, which are hopefully easy for the authors to take into account in the revision.

1. Introduction- The author introduced the negative impacts of UHI. However, relevant research on UHI is lacking. Some important relevant references should be cited as follows.

Spatial Variability and Temporal Heterogeneity of Surface Urban Heat Island Patterns and the Suitability of Local Climate Zones for Land Surface Temperature Characterization. DOI: 10.3390/rs13214338.

Influence of urban morphological characteristics on thermal environment, Sustainable Cities and Society (2021), https://doi.org/10.1016/j.scs.2021.103045.

Contribution of urban ventilation to the thermal environment and urban energy demand: Different climate background perspectives, Science of the Total Environment (2021), https://doi.org/10.1016/j.scitotenv.2021.148791.

Suitability of human settlements in mountainous areas from the perspective of ventilation: a case study of the main urban area of Chongqing, Journal of Cleaner Production (2021), https://doi.org/10.1016/j.jclepro.2021.127467.

The impact of urban renewal on land surface temperature changes: A case study in the main city of Guangzhou, China. Remote Sensing (2020), https://doi.org/10.3390/rs12050794.

Impacts of Neighboring Buildings on the Cold Island Effect of Central Parks: A Case Study of Beijing, China. Sustainability (2020), doi: 10.3390/su12229499.

Additional detail regarding the influence of urban morphology on urban temperatures has been included and the references listed have now been included. 

2. The author emphasizes UHI, while temperature was chosen in this manuscript. 27 ˚C, 33˚C and 37 ˚C can represent UHI? It is more appropriate to explore the effects of temperature.

The rational for using specific temperatures has been included with appropriate citation (introduction paragraph 3). Additional detail has been added to the methods section (methods; page 7). 

3. Figure 2 is not clear enough, please modify it.

Figure 2 has been modified for clarity, presenting the results for each temperature as separate graphs and indicating the location of the pre-molt peak that occurs 2 days prior to molt 2.

4. There is no Conclusion in this manuscript, please confirm it.

The last paragraph of the discussion has been reformatted to more concisely summarize key findings as a Conclusion Section.

Reviewer 2

After a very careful reading of the work entitled "Ecdysteroid Responses to Urban Heat Island Conditions during Development of the Western Black Widow Spider", I have found a very well-done work, well presented, and organized, clear in concepts and methodology. The topic and context attract attention for many readers from various disciplines. The study is worth to be published after conducting the revisions.

I just suggest the author(s) add uncertainty analysis to enhance the scientific rigor of the research.

This paper addresses the question of how ecdysteroid measurements recorded from spiderlings reared at different temperatures compare to one another rather than addressing the absolute uncertainties of ecdysteroid measurement techniques. We therefore give results with statistical uncertainties (mean, standard error), rather than completing additional uncertainty analysis, which we believe is outside the scope of this paper.

Additional Comments:

All figures were converted using PACE to ensure compliance with PLOS requirements.

---

## [Decision Letter · Decision Letter 1]

8 Apr 2022

Ecdysteroid responses to urban heat island conditions during development of the western black widow spider (Latrodectus hesperus)

PONE-D-21-35850R1

Dear Dr. Hackney Price,

We’re pleased to inform you that your manuscript has been judged scientifically suitable for publication and will be formally accepted for publication once it meets all outstanding technical requirements.

Kind regards,

Jun Yang

Academic Editor

PLOS ONE

Additional Editor Comments (optional):

Accept

Reviewers' comments:

Reviewer's Responses to Questions

**Comments to the Author**

1. If the authors have adequately addressed your comments raised in a previous round of review and you feel that this manuscript is now acceptable for publication, you may indicate that here to bypass the “Comments to the Author” section, enter your conflict of interest statement in the “Confidential to Editor” section, and submit your "Accept" recommendation.

Reviewer #1: All comments have been addressed

Reviewer #2: All comments have been addressed

2. Is the manuscript technically sound, and do the data support the conclusions?

Reviewer #1: Yes

Reviewer #2: Yes

3. Has the statistical analysis been performed appropriately and rigorously? 

Reviewer #1: Yes

Reviewer #2: Yes

4. Have the authors made all data underlying the findings in their manuscript fully available?

Reviewer #1: Yes

Reviewer #2: Yes

5. Is the manuscript presented in an intelligible fashion and written in standard English?

Reviewer #1: Yes

Reviewer #2: Yes

6. Review Comments to the Author

Reviewer #1: (No Response)

Reviewer #2: This paper had been revised according to the comments of the reviewers. I have no further comments. The current version of the paper is worth publishing.

7. PLOS authors have the option to publish the peer review history of their article (what does this mean?). If published, this will include your full peer review and any attached files.

Reviewer #1: No

Reviewer #2: No

---

## [Editor Report · Acceptance letter]

20 Apr 2022

PONE-D-21-35850R1 

Ecdysteroid responses to urban heat island conditions during development of the western black widow spider (*Latrodectus hesperus*) 

Dear Dr. Hackney Price:

I'm pleased to inform you that your manuscript has been deemed suitable for publication in PLOS ONE. Congratulations! Your manuscript is now with our production department. 

Kind regards, 

on behalf of

Dr. Jun Yang 

Academic Editor

PLOS ONE